# The Emerging Role of Extracellular Vesicles Detected in Different Biological Fluids in COPD

**DOI:** 10.3390/ijms23095136

**Published:** 2022-05-05

**Authors:** Tommaso Neri, Alessandro Celi, Mariaenrica Tinè, Nicol Bernardinello, Manuel G. Cosio, Marina Saetta, Dario Nieri, Erica Bazzan

**Affiliations:** 1Centro Dipartimentale di Biologia Cellulare Cardiorespiratoria, Dipartimento di Patologia Chirurgica, Medica, Molecolare e dell’Area Critica, Università degli Studi di Pisa, 56124 Pisa, Italy; tommaso.neri79@for.unipi.it (T.N.); alessandro.celi@unipi.it (A.C.); dario.nieri@ao-pisa.toscana.it (D.N.); 2Department of Cardiac, Thoracic, Vascular Sciences and Public Health, University of Padova, 35128 Padova, Italy; mariaenrica.tine@unipd.it (M.T.); nicol.bernardinello@unipd.it (N.B.); manuel.cosio@mcgill.ca (M.G.C.); marina.saetta@unipd.it (M.S.); 3Meakins-Christie Laboratories, Respiratory Division, McGill University, Montreal, QC H3A 0G4, Canada

**Keywords:** extracellular vesicles, chronic obstructive pulmonary disease, pathogenesis, biomarkers

## Abstract

The pathogenesis of chronic obstructive pulmonary disease (COPD) is characterized by complex cellular and molecular mechanisms, not fully elucidated so far. It involves inflammatory cells (monocytes/macrophages, neutrophils, lymphocytes), cytokines, chemokines and, probably, new players yet to be clearly identified and described. Chronic local and systemic inflammation, lung aging and cellular senescence are key pathological events in COPD development and progression over time. Extracellular vesicles (EVs), released by virtually all cells both as microvesicles and exosomes into different biological fluids, are involved in intercellular communication and, therefore, represent intriguing players in pathobiological mechanisms (including those characterizing aging and chronic diseases); moreover, the role of EVs as biomarkers in different diseases, including COPD, is rapidly gaining recognition. In this review, after recalling the essential steps of COPD pathogenesis, we summarize the current evidence on the roles of EVs collected in different biological mediums as biomarkers in COPD and as potential players in the specific mechanisms leading to disease development. We will also briefly review the data on EV as potential therapeutic targets and potential therapeutic agents.

## 1. Introduction

Chronic obstructive pulmonary disease (COPD) is a chronic, complex, and heterogeneous condition, which develops within several years with different mechanisms [1]. COPD is the third leading cause of mortality worldwide and represents a major health issue, with a significant socioeconomic burden, primarily for (but not limited to) the health care systems [2]. The physiological hallmark of the disease, currently required for diagnosis confirmation, is the irreversible airflow obstruction, and patients often complain about shortness of breath (dyspnea), cough, exercise limitation. Periodically, patients experience acute worsening of these symptoms, the so-called acute exacerbations that can sometimes require hospitalization [3]. Since the landmark studies by Fletcher and Peto on airflow obstruction in COPD [4], a large amount of literature has been published about clinical history, lung function and pathogenesis of COPD. Nonetheless, it is currently clear that COPD is characterized by different biological pathways (called endotypes), not mutually exclusive and with not linear interactions among each other, leading to the development of specific aspects of the disease, which are finally responsible for specific clinical characteristics (phenotypes). However, most of these mechanisms are still debated and under investigation [5]. Significant comorbidities, also referred to as extrapulmonary manifestations [6], play an important role in both management and prognosis of COPD. Finally, COPD is thought to be characterized by an accelerated lung aging, since most hallmarks of aging can be found in patients affected by this disease [7].

Extracellular vesicles (EVs) are submicron particles released into biological fluids (including blood, saliva, sputum, tears, cerebrospinal fluid) by virtually all cells. EVs are involved in cell-to-cell communication in numerous conditions, including aging and cellular senescence processes where they probably represent a form of altered intercellular communication. Even though their nomenclature is still debated [8], the main EVs subtypes currently considered include exosomes (endosome-originated particles) and the larger plasma membrane-derived microvesicles (MVs, formerly called “microparticles”) [9]. EVs have been studied in COPD, mainly as potential biomarkers, aiming to identify specific endo-phenotypes of the disease, including extrapulmonary manifestations [10]. Their biological potential in COPD pathogenesis is currently under investigation.

This is a narrative review, focusing on the role of EVs in COPD. After recalling the essential pathways of the pathogenesis of COPD, we will analyze the current evidence about the suggested role of EVs as biomarkers of the different aspects of the disease (inflammation, pulmonary vascular apoptosis, cardiovascular manifestations) and the available data about their possible involvement in the pathogenesis of the disease, even though data directly obtained on from COPD patients are lacking. Since they can be found in different mediums, we will summarize current evidence on EVs in COPD according to the main biological fluids currently studied in respiratory medicine. We will present an overview of the most significant available data about cellular and animal models involving EVs in the pathogenesis of COPD. Finally, we will review the studies that describe EVs as potential therapeutic targets as well as those in which EVs have been investigated as therapeutic agents.

Since many studies have been already conducted before the currently recommended nomenclature has been proposed [8], we have decided to use the generic term “extracellular vesicles”, and specified the vesicle type (microvesicles, exosomes) when deemed appropriate.

## 2. Pathogenesis of COPD

A significant inflammation is involved in the pathogenesis of COPD where noxious particles (cigarette smoke, biomass dusts, pollutants) induce peripheral airways epithelium release of damage-associated molecular patterns (DAMP) and other danger signals [11], thus activating an innate immune response (via Toll-like receptors and cytokines production) by recruiting macrophages, monocytes, and neutrophils [6]. Later, there is an activation of adaptive immune cells: both T and B-lymphocytes number in the peripheral airways increases, and they often organize into lymphoid follicles that can be observed in lung biopsies of COPD patients. A progressive increase of inflammatory cells (belonging both to innate and adaptive immune response) has been observed, along with disease severity progression [7]. Indeed, a complex crosstalk between innate or adaptive immunologic cells (macrophages, neutrophils, CD4^+^ and CD8^+^ lymphocytes) and other cell types like dendritic or epithelial cells has been described, and it is conceivable that different degrees of immunological activation are involved into COPD pathogenesis [12]. Anyway, although specific genetic susceptibility loci have been identified by genome wide association studies [13], some of the main determinants of the immune response development in COPD, leading to different degrees of disease severity, are still largely unknown [12]. However, the infiltration of the peripheral airway mucosa by the abovementioned inflammatory cells is responsible for the so-called small airways disease, an event clearly related to COPD development, as described by the classical study by Hogg et al. [14]. In this condition, histologic modifications have been observed in small conducting airways, with an increased volume of all the main elements represented in the airway architecture: epithelium, lamina propria, smooth muscle cells and adventitia [14]. Along with small airways disease, inflammatory and innate immune cells also produce lung tissue destruction, mainly through proteases and reactive oxygen species (ROS) release, leading to pulmonary emphysema [6]. Emphysema is characterized by an increase in both alveolar and pulmonary endothelial cells apoptosis [15] with a reduction in vascular endothelial growth factor (VEGF) in these patients, both in sputum and in blood [16]. Interestingly, some experimental data have shown that airway epithelial cells can produce, when exposed to cigarette smoke extract (CSE), pro-inflammatory cytokines like interleukin (IL)-1, which in turn stimulates matrix fibroblast to release IL-8, a potent chemoattractant for neutrophils, thus confirming the complex cellular interplay underlying COPD development [17]. Extracellular matrix (ECM) remodeling play a role in COPD pathogenesis (both in small airways disease and in emphysema), and there are some evidences that an excessive ECM turnover is present in COPD, with increasing circulating levels of ECM components (i.e.,: collagen fibers, elastin fragments) mostly during acute exacerbations [18].

Since the prevalence of COPD is markedly increased in the elderly, the accelerated lung function decline, characteristic of COPD, has been interpreted as an accelerated lung aging [19]. Most of the proposed hallmarks of aging (genomic instability, telomere attrition, epigenetic alterations, loss of proteostasis–autophagy–, deregulated nutrient sensing, mitochondrial dysfunction, cellular senescence, stem cell exhaustion, and altered intercellular communication) [20] can be found in COPD. In particular, studies have shown accelerated telomere length loss [21], increased autophagy [22], accumulation of ubiquinated proteins [23], aberrant proteostasis [24], impairment in DNA repair after damage by ROS [25]. Among these mechanisms, cellular senescence plays a significant role in COPD pathogenesis [26]. Cellular senescence is characterized by a stable cell cycle arrest, and it may develop because of a progressive shortening of telomeres after repeated cellular divisions, thus representing a physiological adaptation to prevent DNA damage (“replicative senescence”). However, an accelerated senescence, independent from telomeres attrition, can be observed as a reaction to genotoxic insults, like oxidative or metabolic stress [27]. Although their replicative cycle is arrested, senescent cells are still metabolically active secreting many factors (cytokines, chemokines, growth factors, proteases), together known as the senescence-associated secretory phenotype (SASP). The SASP response involves MAP kinase p38, which in turn triggers nuclear factor-κB (NF-κB) activation [28]. Through these several pro-inflammatory molecules, included in the SASP, senescent cells may affect bystander cells, thus being involved in the pathogenesis of several diseases, like cancer and age-related chronic disease, including COPD. This pathological process requires communication mediators: interestingly, EVs have been recently explored as active players in SASP [29]. Indeed, EVs, and particularly microvesicles, can cargo, both on their outer membrane and within their cytosol, several active molecules, including messenger RNA (mRNA), microRNAs (miRNAs), intracellular enzymes, mediators of intercellular adhesion, major histocompatibility complex receptors, tissue factor [9]. Since they play a pivotal role in both paracrine and non-hormonal, endocrine intercellular communication, through these biologically active cargoes (which can product different effects in recipient cells), EVs have been implied in senescence and in age-related diseases, as a relevant component of the SASP [30]. Moreover, EVs are released after inflammatory or septic insult, epithelial (including endothelial) activation or cellular apoptosis in respiratory diseases and are therefore extensively studied as potential biomarkers in these conditions, especially in COPD and lung cancer [10].

In this light, their potential as diagnostic tools for precision medicine purposes are intriguing, even though their use in clinical routine is still far away.

## 3. EVs in COPD

### 3.1. Circulating Extracellular Vesicles as Biomarkers in Peripheral Blood

Peripheral venous blood is an easily accessible biological fluid, and it has therefore been extensively sampled to study extracellular vesicles in COPD.

Most studies have focused on the potential value of EVs as biomarkers of specific aspects (both clinical and pathological) of COPD. EVs have been proposed as markers of apoptosis, local and systemic inflammation, disease severity and activity, and as tools to identify specific subgroups of patients (phenotypes).

As already said, pulmonary capillary bed apoptosis is a key event in the pathogenesis of COPD, and specifically of pulmonary emphysema. In their landmark study, Gordon et al. first demonstrated a significant increase in endothelial-derived EVs (EEVs) in smokers with functional markers of pulmonary emphysema (i.e.,: reduced lung diffusion capacity for carbon monoxide, DLCO, measured by pulmonary function tests, PFTs), with respect to smokers with normal PFTs or non-smokers; moreover, EEVs were mainly CD31^+^ (a marker of apoptotic endothelial cells), while CD62E antigen (E selectin, a marker of “activated” endothelium) was less represented on EEVs surface in these subjects, and most EEVs were angiotensin-converting enzyme (ACE) positive, thus implying their genesis from the pulmonary capillary bed. Interestingly, all the studied subjects had normal spirometry, and therefore they had not developed COPD: the Authors concluded that apoptotic, endothelial-derived EVs could represent a reliable marker of early lung tissue (including blood vessels) destruction, a pathological hallmark of pulmonary emphysema that might eventually result in COPD development [31]. In a subsequent study, the same group confirmed these results in patients with an established diagnosis of COPD demonstrating that in these patients circulating CD31^+^ EEVs levels were increased compared to smokers without COPD or non-smokers. Most of EEVs were ACE^+^, confirming their pulmonary capillaries origin. Furthermore, COPD patients showed persistent elevated levels in CD31^+^ EEVs even 12 months after quitting smoking, while in healthy smokers their levels significantly decreased after smoking cessation: these data, together with the previous study, thus confirm that pulmonary endothelial apoptosis likely plays a role in COPD pathogenesis, both at the beginning of this process and in its progression, irrespective of the persistence of the external stimulus (cigarette smoke) [32]. Indeed, in a work by Garcia-Lucio et al., apoptotic (CD31^+^) EEVs were increased in COPD patients compared to non-smokers; moreover, these patients showed reduced levels of circulating endothelial progenitor cells, thus indicating that COPD is characterized by both a significant vascular damage and a reduced cellular regenerating capacity [33]. All these data together are consistent with an accelerated aging process at least of endothelial cells in COPD.

Vascular endothelium is currently considered an important player in COPD, and a “vascular” endo-phenotype of the disease has been proposed [34], also responsible for cardiovascular comorbidities [35]. Tomashow et al. found different profiles of EEVs expression in COPD patients: in particular, increased CD31^+^ EEVs (“apoptotic endothelium”) in emphysema with mild airflow obstruction, and predominant CD62E^+^ EEVs (“activated endothelium”) in severe obstruction with hyperinflation (i.e., pathological gas entrapment in the lung, leading to mechanical impairment in lung emptying during physiological expiration, and thus worsening dyspnea) [36]. These data suggest that different subtypes of EEVs could represent biomarkers of different endotypes in COPD. Interestingly, circulating CD62E^+^ EEVs were also elevated during COPD exacerbations (which are known to be characterized by a significant bout of both local and systemic inflammation [6], but they returned at the pre-exacerbation levels after four weeks, coherently with the reduction in the acute inflammation. However, in the same study, apoptotic CD31^+^ EEVs showed, along with a peak during exacerbation, the persistence of elevated level in the blood after CD62E^+^ EEVs normalization, thus suggesting that the vascular damage caused by the acute exacerbation was still ongoing, even after the acute bout of inflammation was resolved [37]. A relationship between circulating EVs and inflammation has been found also in other studies: in particular, both endothelial-derived EVs [38] and circulating exosomes [39] were significantly and positively correlated with blood levels of interleukin-6 (IL-6), that is involved in atherothrombotic risk both in general population and in COPD patients, in which IL-6 represents a well-known marker of systemic inflammation [40,41]. These data further underline the close interplay between inflammation and vascular damage, also in COPD; in this light, EVs could represent a relevant biological link, and thus deserve deeper investigations [36].

Both circulating endothelium- and monocyte-derived EVs levels progressively increased along with disease severity, evaluated by using airflow obstruction, symptoms and exacerbations history together in a single panel [38]; moreover, CD62E^+^ EEVs showed a significant correlation with lung function decline over time [42] and with exacerbation susceptibility [35]. Finally, EEVs have also been proposed as markers of endothelial function under stress conditions (such as physical exercise), even though with discordant results [43,44].

Among the several possible cargoes included within EVs, miRNAs probably represent one of the most significant. Some studies have isolated specific miRNAs in circulating EVs in COPD patients. In a small study, Sundar et al. found that COPD patients had a specific miRNAs subset within circulating EVs (mostly exosomes), different from those of smokers without COPD or non-smokers [45]. Similar results have been obtained in another study, that demonstrated a significant increase in a specific miRNA (miR-21) content within circulating exosomes in COPD patients, compared to smokers without COPD [46]. Finally, Carpi et al. found that some muscle-specific miRNAs (miR-206, miR-133a-5p and miR-133a-3p), included in circulating medium-large EV, were up-regulated in a subgroup of COPD patients, characterized by a significant amount of respiratory symptoms [47]. These results suggest that EV-embedded miRNAs could represent a sort of signature to identify COPD patients, and even specific subgroups among them.

Taken together, these data show the growing importance of circulating EVs in the blood as potential biomarkers for different aspects of COPD, involved in both disease development and clinical expression: endothelial apoptosis, systemic inflammation, lung function decline over time and exacerbations susceptibility. In particular, the several studies on endothelial-derived EVs (EEVs) showed coherent results, thus suggesting that EEVs could represent not only a reliable marker of specific domains of COPD, but also a potential player in the development of disease. Anyway, although attractive, circulating EVs are not still used in clinical routine for COPD patients management, and further research is needed to fill this gap.

### 3.2. Extracellular Vesicles as Biomarkers in Bronchoalveolar Lavage Fluid (BALF) and Sputum

The presence of EVs in tissue fluids, such as BALF from the lung, offers the unique possibility of focusing specifically on the presence, origin, and roles of these vesicles. The study of BALF has the advantage of identifying the cell origin of EVs beyond the cargo, an important distinction that might increase the prognostic and possible mechanistic value of EVs in the study of lung events in COPD.

BALF, a fluid collected during bronchoscopy, is an important diagnostic material that provides information about different inflammatory processes taking place in the alveolar space. It is enriched with various cellular and non-cellular content (epithelial cells, macrophages, cytokines, extracellular vesicles) and represents an ideal “liquid biopsy” source of biomarkers for different lung diseases.

BALF contains a substantial amount of different EVs, although in a lower concentration than in other biological fluids, such as plasma. Although studying BALF is potentially very important for understanding the pathogenesis of COPD, to date only few studies have evaluated the EVs in BALF.

Rodriguez et al. for the first time compared the concentrations of EVs isolated from plasma and BALF simultaneously, and they conclude that EVs levels were significantly higher in plasma than in BALF samples in both groups of patients (1.8 and 3.8 × 10^8^ EVs-like particles in BALF vs. 4.0–9.8 × 10^8^ particles per mL in plasma) [48]. Similarly, Zabera et al., in unpublished data, detected higher particles concentrations in plasma than in BALF: nevertheless, the Authors stated that the majority of detected particles in BALF are “true” EVs, whereas, in plasma, most particles represent non-EVs structures such as lipoproteins [49].

In BALF macrophages are considered to be major sources of pulmonary EVs, which regulate the normal airway biology including homeostasis and innate defense [49,50]. The BALF-EVs composition may change dramatically in disease.

Genschmer et al. identified and characterized activated neutrophil-derived exosomes (CD66b^+^/CD63^+^) in the BALF of both COPD patients and normal controls. While the number of such exosomes was similar in both groups, only exosomes of COPD patients stained positive for neutrophil elastase (NE). Based on this observation, the Author hypothesized a pathogenetic role for these structures that will be discussed in detail below [51].

The importance of neutrophil-derived EVs was also confirmed in a clinical study conducted by Soni et al., who analyzed different microvesicles (MVs) populations in BALF from patients with mild to very severe COPD, and successfully identified various MVs subtype populations. Neutrophil-derived MVs were the only population that are correlated with a number of key functional and clinically relevant disease severity indexes, suggesting the potential of BALF neutrophilic MVs as a novel COPD biomarker, that tightly links a key pathophysiological mechanism of COPD (intra-alveolar neutrophils activation) with clinical severity/outcome (Figure 1) [52].

These elegant studies support the need for further studies on the association between proteases and extracellular vesicles from other cell types, such as macrophages, in the pathogenesis of COPD. The pioneering study by our group recently described the presence and source of MVs in BALF of smokers with and without COPD, compared with nonsmoking controls [53]. The results showed a peculiar increase of macrophage-derived MVs in the lungs of smokers with COPD compared to smokers without airflow limitation and to healthy controls. Macrophage-derived MVs numbers correlated with both smoking exposure and degree of airway obstruction. In our cohort, BALF MVs derived from endothelial cells were similarly represented regardless of the presence of COPD or smoking history, in contrast with peripheral endothelial-derived MVs reported to increase in patients with COPD and smokers. These findings support the known inflammatory role of alveolar macrophages in COPD and the potential contribution of the lung capillary endothelial cells in enhancing this inflammation. These results open the opportunity for future investigation of these EVs as biomarkers and possible mechanistic guides in COPD.

At the same time, Qiu et al. described that the levels of both CD4^+^ and CD8^+^ T-lymphocyte-derived extracellular vesicles were remarkably elevated in BALF of COPD patients, which was consistent with the increase of EV-associated T lymphocytes in the development of COPD [54].

Increased levels of EVs derived from T cell could be caused by the activation of CD4^+^/CD8^+^ T lymphocytes that already exist in the airways, or by the accumulation of up-regulated CD4^+^/CD8^+^ cells, recruited during the development of COPD, suggesting that also these subtypes of EVs could be considered putative biomarkers to distinguish different types of inflammation.

Induced sputum has been suggested as a non-invasive surrogate of BALF for the identification of biomarkers in daily clinical routine [55]; nevertheless, our knowledge on EVs in this medium is still too immature to consider them a reliable biomarker, since most of sputum fluid derives from upper airways. To our knowledge, only one study has evaluated EVs in sputum from COPD patients [56]: the goal of this study is to investigate the presence and source of sputum EVs in COPD patients and to correlate them to the clinical picture. The main result is not only the presence of EVs in COPD patients’ sputum, but also the relation between the number of endothelial EVs and Forced Expiratory Volume in the 1^st^ second (FEV1, the main pulmonary function index in COPD), thus indicating that endothelial injury is closely connected to the pathophysiology of COPD.

As well as for blood, miRNAs represent one of the most significant cargos of EVs also in BALF, especially for their role in epigenetic regulation. Over these last years, many studies have tried to shed light on the role of miRNAs in the COPD pathogenesis. Most of them analyzed miRNAs derived from exosomes and showed that they were significantly altered in COPD patients compared to healthy subjects. Armstrong et al. found an altered expression of miR-451 and miR-663, two miRNAs acting on matrix metalloproteinase and TGF-β1 [57].

In their notably innovative study, Kaur et al. determined and compared the miRNA profiles of BALF-derived exosomes of healthy non-smokers, smokers, and COPD patients using next generation sequencing. The results of this study are surprising, since there is not a different miRNAs profile in the BALF-derived exosomes from healthy smokers and non-smokers, suggesting that smoking status alone does not affect the exosome-mediated signaling in healthy individuals. However, the Authors found a distinct variation in the miRNAs populations between COPD patients and healthy non-smokers: in particular, two miRNAs (miR-423-5p and miR-100-5p), involved in regulation of apoptosis, are down-regulated, while miR-320b and miR-22-3p (involved in the emphysema control) are up-regulated [58].

Again, as for circulating EVs in peripheral blood, EVs in BALF represent an attractive opportunity to identify both novel biomarkers and potential players in disease pathogenesis. However, the reported data are still preliminary; moreover, BALF sampling is obviously more invasive than blood collection, and this could reasonably limit its potential as a source of data.

Figure 1 summarized the EVs released by different cell types in alveolus and blood. 

### 3.3. Extracellular Vesicles in the Pathogenesis of COPD and in COPD Models

Several groups have investigated the involvement of EVs in the pathogenesis of COPD, both in in vitro (by using cellular lines), or in vivo (through the use of animal models) studies.

Since cigarette smoke has classically been considered a main determinant of COPD development, in particular as a pivotal trigger for lung inflammation and its consequences (see the “pathogenesis” section), many studies have explored EVs role in this model. Our group has shown that mononuclear cells (whose involvement in the first steps of lung inflammatory damage in COPD is well known) shed EVs upon stimulation with CSE. Interestingly, these EVs cause upregulation of proinflammatory mediator synthesis (interleukin-8, intercellular adhesion molecule-1 and C-C motive chemokine ligand 2) by lung epithelial cells, thus providing a possible mechanism for further recruitment of innate immune cells (neutrophils and monocytes) in the site of inflammation [59].

Others papers focused on the role of EVs in pathological changes in airway structures due to exposure to CSE. In the study by Fujita et al. a miRNAs/EV-mediated cellular communication mechanism, between bronchial epithelial cells and lung fibroblasts, promotes the development of smoke-induced myofibroblast differentiation. According to the Authors, this novel mechanism of myofibroblast differentiation from lung fibroblasts is attributed to miR-210, carried by EVs (released by bronchial epithelial cells upon CSE stimulus) and regulating the autophagy machinery [60]. Xu et al. demonstrated that CSE induced the up-regulation of miR-21 levels in exosomes (derived from human bronchial epithelial cells) and promoted myofibroblast differentiation through increased hypoxia-inducible factor 1α transcriptional activity [46]. These data clearly lend support to an active role of EVs in airways remodeling, thus contributing to the small airways disease in COPD. On the other hand, in an interesting study by He et al., BEAS-2B cells were shown to indirectly modulate the epithelial–mesenchymal transition process in COPD pathogenesis, by alleviating the polarization of M2 macrophages via reducing their EVs miR-21 content, thus revealing a novel purported compensatory role of EVs in COPD [61]. Last, Wang et al. demonstrated, in a murine model, that exosomes derived from CSE-treated airway epithelial cells promote M1 macrophage polarization through the upregulation of triggering receptor expressed on myeloid cells-1 expression and aggravate CSE-induced impairment in pulmonary function and lung injury, thus suggesting that the M1 polarization induced by EVs might be one of the mechanisms by which smoking promotes the progression of COPD [62].

Other studies have focused on the role of EVs in pathological changes in airway structures by using CSE-unrelated models. Genschmer and co-authors demonstrated that neutrophils release EVs that stain positive for CD63 (a marker of endosomal origin) and were therefore described as exosomes. Upon activation with the formylated peptide, f-MLP, the number of exosomes did not increase; however, only exosomes from activated neutrophils stained positive for NE. Moreover, exosome-bound NE was resistant to α1-antitrypsin. These NE-bearing EV were capable of degrading extracellular matrix and, when instilled intratracheally, to induce the development of emphysema in mice. The Authors also demonstrated that CD63^+^/CD66b^+^ EVs (exosomes of neutrophilic origin) from the BAL of COPD patients were able to induce emphysema in mice, while a similar number of EVs from normal subjects did not. Taken together, the data are consistent with a model whereby neutrophil-derived exosomes capture NE from degranulated neutrophils, protect it from α1-antitrypsin and contribute to the development of emphysema [48], which represents, together with the small airways disease, the main pathological feature of COPD [15].

Because of the pivotal role of NE in lung tissue destruction (and thus in emphysema development), other studies explored this mechanism. Margaroli et al. have demonstrated that EVs from in vivo LPS-activated mouse neutrophils induced a COPD-like disease in naive recipients, through a NE-dependent mechanism [63]. Similarly, other Authors collected tracheal exosomes aspirated from both preterm patients with bronchopulmonary dysplasia (BPD) and controls born at term, and subsequently treated saccular stage mouse lung fibroblasts with exosomes clustered from BPD or control patients. They show that exosomes from BPD patients had higher NE and were able to down-regulate two fibroblastic proteins involved in elastic fiber development, thus hypothesizing that NE-containing exosomes were involved in blocking the production of critical elastic fiber assembly components [64].

The role of pulmonary infection in COPD development is well known [3], but evidence on EVs role in infectious events in COPD is scarce. A role for bacteria-derived EV has been described in the pathogenesis of pneumonia (reviewed in [65]). In vitro studies have shown, for example, that S. Pneumoniae produces an extracellular vesicle-associated endodeoxyrybonuclease that degrades neutrophil extracellular traps, thus contributing to the evasion of the innate immune system [66]. Since as many as 80% of acute exacerbations of COPD are triggered by bacteria or viruses [67], it is conceivable that bacteria- and virus-derived EV might contribute to their pathogenesis, even though no direct data are available.

All these studies are quite intriguing about the active role of EVs in COPD pathogenesis, and strongly underline their pivotal importance in cell-to-cell communication. Of course, since these results have been obtained in cellular or animal models, they cannot be uncritically translated to the biology of human COPD. In this light, studies directly conducted on COPD patients are needed to firmly elucidate EVs biological functions in the disease development.

### 3.4. EVs as Potential Targets for Treatment

Based on the known role of EV in the pathogenesis of different diseases, researchers have been investigating different molecules that might function as inhibitors of EV release [68]. Relevant to COPD, we have demonstrated, in an in vitro model, that tiotropium, a drug routinely used as maintenance therapy in COPD, was able to inhibit EVs generation (induced by acetylcholine) by bronchial and endothelial cells [69]. This hitherto unrecognized effect of tiotropium could help explain its therapeutic effect in reducing COPD acute exacerbations, also characterized by higher EVs concentrations [37].

### 3.5. Potential Therapeutic Effects of EVs in COPD

The use of EVs as therapeutic agents in different conditions is being actively investigated [70]. Several groups have evaluated the potential use of EVs derived from mesenchymal stem cells in COPD. Ridzuan et al. have used a rat model of COPD induced by exposure to cigarette smoke. The administration of mesenchymal cell-derived EVs caused a significant reduction of peribronchial, perivascular and parenchymal inflammation, as well as the loss of alveolar septa. The results were similar to those obtained with mesenchymal stem cells, suggesting that EVs might serve a cell-free-based therapy for COPD [71]. A protective effect of mesenchymal stem cell-derived EV was also observed on cigarette smoke-induced mitochondrial dysfunction in mice [72].

Table 1 provides a summary of the main findings from the most important studies conducted on EVs, both in COPD patients and in cellular or animal models of the disease.

## 4. Conclusions

In the last years, EVs have gained progressive importance in COPD, a complex and heterogeneous disease, with several pathogenic pathways, including lung aging and cellular senescence; indeed, the amount of published studies on the topic has produced a significant evolution in the knowledge in this area.

First, there is a growing body of evidence about EVs role as potential diagnostic and prognostic biomarkers in COPD, useful to identify specific endo-phenotypes of the disease. In this light, plasma and serum are most popular sources of EVs, as they are easily accessible, while BALF is more difficult to obtain, due to the invasiveness of its collection, although it reasonably better represents airway inflammation and lung microenvironment.

Moreover, EVs are currently under investigation as potential players in COPD pathogenesis, both using in vitro and in vivo models of the disease, with promising results which are beginning to shed light on the complex mechanisms underlying COPD development and progression. Indeed, some of the biological cargoes included into EVs, like miRNAs, are considered to be potential modulators of biological processes in terms of intercellular communication.

However, a complete picture of EVs active functions in COPD is still lacking: a better understanding of the role of EVs both in the inflammatory lung microenvironment and in the systemic circulation is therefore needed, to improve our knowledge about the mechanisms underlying the disease, to be subsequently translated into clinical practice, for an effective and personalized management of COPD patients.

## Figures and Tables

**Figure 1 ijms-23-05136-f001:**
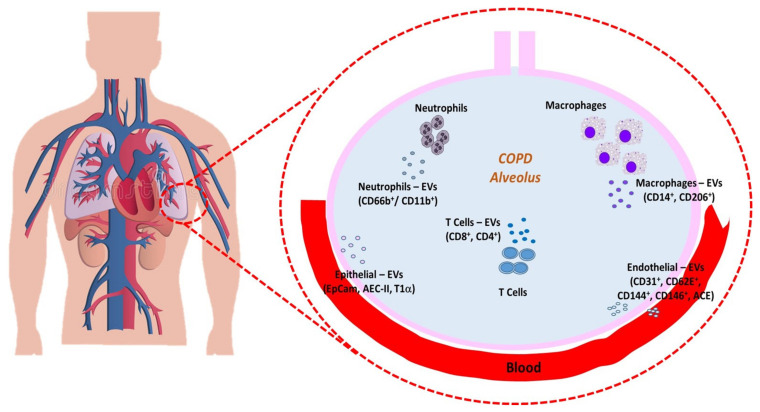
Summary of the EVs released from almost cell types in alveolar space and blood. EVs play a key cell-to-cell communicator role in the lung microenvironment and in the COPD pathogenesis.

**Table 1 ijms-23-05136-t001:** Main findings about EVs in patients affected by COPD and in cellular or animal models if the disease.

Significance or Hypothesised Role for EVs	Source	Medium	Studied Population	References
Markers of endothelial apoptosis	Endothelium	Peripheral blood	COPD patients	[31,32,33,36,37]
Markers of endothelial dysfunction under stress	Endothelium	Peripheral blood	COPD patients	[43,44]
Markers of disease severity	Endothelium Monocytes Neutrophils Endothelium	Peripheral blood BALF sputum	COPD patients	[38,42] [52,53] [56]
Markers of inflammation	Endothelium Monocytes Lymphocytes	Peripheral blood BALF	COPD patients	[36,37,38,39] [54]
EV-miRNAs induce remodelling (myofibroblast transition from lung fibroblast)	n.a.	n.a. Peripheral blood	In vitro model Mixed model (human/murine)	[60] [46]
EVs induce emphysema-like disease	neutrophils	BALF	Mixed model (human/murine) Murine model	[51,64] [63]
Protective role for EVs against inflammation	Mesenchymal cells	n.a. n.a.	In vitro model Murine model	[71] [72]
EVs as target of specific COPD treatments	Bronchial cells Endothelium	n.a.	In vitro model	[69]

EVs: extracellular vesicles; COPD: chronic obstructive pulmonary disease; BALF: bronchoalveolar lavage fluid; n.a.: not applicable.

## Data Availability

Not applicable.

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
