# Peer review of "The Emerging Role of Extracellular Vesicles Detected in Different Biological Fluids in COPD"

_ijms, 2022, doi:10.3390/ijms23095136_

Round 1

Reviewer 1 Report

This review article was clear, generally structured logically and covered an area of interest. The main improvements I felt could be made were structural rather than content focussed

  1. Introduction - the opening section flowed well but I was not sure what the section  1.1 was supposed to achieve? Up to line 115 or so it was just a refresher on COPD pathogenesis, and after that it seemed to try and link EVs to COPD. If this was the intention then a second subheading (eg 1.2 How are EVs linked to COPD pathogenesis?) might be appropriate. As is it seemed odd to have a section 1.1 but no others.
  2. Body text - this was comprehensive and the sections made sense. Table 1 was helpful. However I would have appreciated a diagram or two to emphasise the origins of EVs in COPD and/or their relationships with clinical/demographic features and/or their surface markers and potential role in treatment for the future. Review articles can be quite dense in such specialist areas unless they are broken up with images.

Author Response

Reviewer 1

This review article was clear, generally structured logically and covered an area of interest. The main improvements I felt could be made were structural rather than content focused.

We thank the Reviewer for appreciating the Review.

Q1R1: Introduction - the opening section flowed well but I was not sure what the section  1.1 was supposed to achieve? Up to line 115 or so it was just a refresher on COPD pathogenesis, and after that it seemed to try and link EVs to COPD. If this was the intention then a second subheading (eg 1.2 How are EVs linked to COPD pathogenesis?) might be appropriate. As is it seemed odd to have a section 1.1 but no others.

R1R1: As suggested by the reviewer, we have reorganized the paragraphs as follows: 1. Introduction, 2. Pathogenesis of COPD and 3. EVs in COPD.

Q2R1: Body text - this was comprehensive and the sections made sense. Table 1 was helpful. However I would have appreciated a diagram or two to emphasise the origins of EVs in COPD and/or their relationships with clinical/demographic features and/or their surface markers and potential role in treatment for the future. Review articles can be quite dense in such specialist areas unless they are broken up with images.

R2R1: We agree with the reviewer that a diagram interrupts the monotony of the text and helps the reading. As suggested by the reviewer, we now added a picture that emphasizes the origins of EVs and their specific surface markers.

Reviewer 2 Report

The authors provide a narrative review on the current knowledge on EVs in COPD, both from a potential biomarker perspective and a mechanistic perspective. This dual perspective is lacking in the title; at this point the mechanistic perspective is highlighted, yet the evidence for this is too limited so far. A clearer distinction should be made in the manuscript between the sections on EVs as biomarkers vs pathogenic mechanisms of EVs.

The manuscript itself is not very original. Other reviews have been published in recent years, including a systematic review (Gomez 2022), on EVs in COPD.

Considering the manuscript is submitted to a special issue on EVs in a journal on basic science, the pathogenesis of COPD is explained rather briefly. Moreover, it heavily emphasizes COPD as a disease of accelerated aging, and does not detail the immunological complexity, nor does it fully describe the extent of tissue/matrix remodeling that is present and the contribution of eg fibroblasts.

From a biomarker perspective it makes sense to start with EVs in blood, yet not for a lung disease per se. There is also no clear mention of the complexity using blood, as the signal (and pathogenic mechanisms) could relate to co-morbidities of COPD rather than the lung disease itself. An example thereof is the reference by Carpi et al describing muscle-specific miRNAs. Overal, the cargo of EVs is not necessarily useful only as a biomarker but could be related to potential pathogenic mechanisms (muscle wasting).

In the section on EVs in BALF it is stated that lung epithelial cells and macrophages are the main sources of pulmonary EVs with regulatory functions. Yet there is not a single reference to EVs from epithelial cells, and there is also more evidence from neutrophil-derived EVs compared to macrophages.  With respect to neutrophils it is not commented on to what extent EVs are superior over neutrophils per se or MPO as biomarker, how feasible the measurements are. Instead of epithelial-derived EVs, endothelial-derived EVs are found in BALF. How do they end up there? If in vitro studies focus on epithelial cells as source of EVs, why are there no clinical studies measuring these?  In aggregate there needs to be a more critical review of the results obtained in clinical samples, as well as a section on feasibility and reliability of the measurements, including of sample storage.

The potential therapeutic use of EVs from stem cells should be put under a separate subsection. This is relevant, yet it is unexpected from the title and abstract.

Lastly, EVs can also derive from bacteria (outer membrane vesicles). It should be considered adding a section on these in relation to the lung microbiome and/or exacerbations.

A graphical representation would be of benefit as would careful editing of some phrases to make their meaning more clear.

Author Response

Reviewer 2

Q1R2: The authors provide a narrative review on the current knowledge on EVs in COPD, both from a potential biomarker perspective and a mechanistic perspective. This dual perspective is lacking in the title at this point the mechanistic perspective is highlighted, yet the evidence for this is too limited so far. A clearer distinction should be made in the manuscript between the sections on EVs as biomarkers vs pathogenic mechanisms of EVs.

R1R2: We certainly agree that the title did not reflect the scope of the text and we changed it accordingly. Furthermore, we changed the subheadings to better differentiate between the paragraphs devoted to the analysis of EV as biomarkers and the paragraphs that describes their potential pathogenetic role and therapeutic use.

Q2R2: The manuscript itself is not very original. Other reviews have been published in recent years, including a systematic review (Gomez 2022), on EVs in COPD.Considering the manuscript is submitted to a special issue on EVs in a journal on basic science, the pathogenesis of COPD is explained rather briefly. Moreover, it heavily emphasizes COPD as a disease of accelerated aging, and does not detail the immunological complexity, nor does it fully describe the extent of tissue/matrix remodeling that is present and the contribution of eg fibroblasts.

R2R2: The topic covered by the manuscript is in continuous evolution, since EV are being investigated by numerous groups and COPD represents a highly prevalent condition.

We have tried to expand the paragraph about COPD pathogenesis according to the Reviewer’s suggestions. We agree with the Reviewer that it is still a not fully comprehensive description of COPD pathogenesis but, since the review is limited in scopus, and mainly devoted to EVs in COPD, we preferred not to investigate further the immunological complexity or the extracellular matrix remodeling (including the fibroblasts role), all aspects that, though scientifically relevant in COPD, we believe are beyond the aim of our manuscript. We preferred to emphasize the accelerated lung aging in COPD, since it represents a main focus of the Special Issue

Q3R2: From a biomarker perspective it makes sense to start with EVs in blood, yet not for a lung disease per se. There is also no clear mention of the complexity using blood, as the signal (and pathogenic mechanisms) could relate to co-morbidities of COPD rather than the lung disease itself. An example thereof is the reference by Carpi et al describing muscle-specific miRNAs. Overal, the cargo of EVs is not necessarily useful only as a biomarker but could be related to potential pathogenic mechanisms (muscle wasting).

According to the Global initiative against Obstructive Lung Disease (GOLD), COPD is characterized by the presence of comorbidities. However, several Authors prefer the term “extrapulmonary manifestations” that better reflects the concept that COPD is a systemic disease (see, for example, Fabbri LM and Rabe K, From COPD to chronic systemic inflammation? The Lancet, 370:797-799, 2007). Furthermore, the vast majority of studies on inflammatory biomarkers of COPD use blood as the source of biologic material (see, for example, Faner R et al., Lessons from ECLIPSE: a review of COPD biomarkers. Thorax, 69:666-672, 2014). With this in mind, we maintain that describing the condition starting with peripheral blood biomarkers is in line with current concepts. We agree that EVs might be involved in pathogenic mechanisms; unfortunately, this remains a hypothesis. As a result, in a study that we have previously conducted (the abovementioned paper by Carpi et al) for example, we could not make such a claim, since the data only allowed us to speculate about the possible pathogenic role of EV-embedded miRNAs, but we could not be sure about it, as stated in the discussion.

Q4R2: In the section on EVs in BALF it is stated that lung epithelial cells and macrophages are the main sources of pulmonary EVs with regulatory functions. Yet there is not a single reference to EVs from epithelial cells, and there is also more evidence from neutrophil-derived EVs compared to macrophages. With respect to neutrophils it is not commented on to what extent EVs are superior over neutrophils per se or MPO as biomarker, how feasible the measurements are. Instead of epithelial-derived EVs, endothelial-derived EVs are found in BALF. How do they end up there? If in vitro studies focus on epithelial cells as source of EVs, why are there no clinical studies measuring these?  In aggregate there needs to be a more critical review of the results obtained in clinical samples, as well as a section on feasibility and reliability of the measurements, including of sample storage.

R4R2: We added reference in the sentence of EVs from epithelial cells. As reported by Gomez (Respiratory Research 2022) different studies (6 to date) observed EVs derived from epithelial cells and only two studies describe EVs from neutrophils.

We agree with the reviewer that also endothelial-derived EVs were found in BAL (as reported by our research group) and they are there because the lung is heavily supplied by capillaries. For its physiological function, the alveolar septa are formed by a very thin barrier with very large surface area between air and blood. This barrier comprises three extremely thin tissue layers: an endothelium lining the capillaries, an epithelium lining the airspaces, and an interstitial layer to house the connective tissue fibers; epithelium and endothelium make up about one-quarter each of the tissue barrier in the alveolar walls. Therefore, due to the particular lung structural conformation, we believe that it is normal to find endothelial-derived EVs.

Some studies on diseases other than COPD have evaluated the presence of epithelial EVs in BAL, however, the aim of our review was to provide a general overview of the studies in the literature on biological fluids in COPD, these are naturally more numerous for the blood and developing for BAL, considering also the ethical issues related to obtain BAL in COPD patients.

Q5R2: The potential therapeutic use of EVs from stem cells should be put under a separate subsection. This is relevant, yet it is unexpected from the title and abstract.

As the reviewer suggests we added two new short paragraphs titled 4 “EVs as potential targets for treatment” and “Potential therapeutic effects of EVs in COPD”.

Q6R2: Lastly, EVs can also derive from bacteria (outer membrane vesicles). It should be considered adding a section on these in relation to the lung microbiome and/or exacerbations.

R6R2: As the reviewer suggests we added a new short paragraph that mentions the potential role of EVs derived from bacteria in acute exacerbations of COPD.

Q7R2: A graphical representation would be of benefit as would careful editing of some phrases to make their meaning more clear.

R7R2: We agree with the reviewer that a graphical representation helps the reading. As suggested by the reviewer, we now added a picture that emphasizes the origins of EVs and their specific surface markers. The text has been carefully revised.

Round 2

Reviewer 2 Report

The authors have sufficiently addressed most comments from the initial review and have improved the writing. 

However, the newly added sections and sentences did not undergo rigorous editing prior to submission. It is especially noted that the authors like to continue sentences after a semicolon with indeed. It would be advised to switch this up with entirely independent sentences that start differently to augment readability.  The Wang paper is referred to and quoted twice, on page 11 and 12. Text is redundant. On page 13, TREM1 as a target has been deleted, yet the reference still stands. Text on page 3-4 should not be bold.

Content wise, EVs (amounts, sources and content) can not only be linked to the senescence hallmark of aging, but represent in essence a form of altered intercellular communication. This should be added.

The suggestion to include some discussion and critical remarks has not been taken up by the authors. Reviewing merely findings provided in literature is not what a review should do, it should provide context and guidance to the readership. I repeat my comment on the use of neutrophil EVs vs other markers of neutrophils - EVs are not easily measured, lots of pitfalls.... Secondly, it should at least be mentioned that although in health epithelial cells represent a major source of EVs, this has not been examined or found in clinical samples in COPD. Can the authors speculate on the reasons? This is especially relevant given that in vitro studies do focus on these cells as source, but also in relation to the spread of senescence from senescent epithelial cells. There is a lack of connection between the clinical evidence and some of the mechanistic studies and conceptual framework that deserves attention.

The addition of the figure is a good start - but it does not include COPD or the blood. Instead it contains info that is not mentioned in the text (markers of cellular origin of EVs, platelets as a source).

Author Response

The authors have sufficiently addressed most comments from the initial review and have improved the writing.

1. However, the newly added sections and sentences did not undergo rigorous editing prior to submission. It is especially noted that the authors like to continue sentences after a semicolon with indeed. It would be advised to switch this up with entirely independent sentences that start differently to augment readability.

Response: we edited the text

2. The Wang paper is referred to and quoted twice, on page 11 and 12. Text is redundant.

Response: we changed the text

3. On page 13, TREM1 as a target has been deleted, yet the reference still stands. Text on page 3-4 should not be bold.

Response: we modified the text by adding the role of TREM1 in the polarization of M1 macrophages. Sorry for the bold text, unfortunately this was not present in the version uploaded by us.

4. Content wise, EVs (amounts, sources and content) can not only be linked to the senescence hallmark of aging, but represent in essence a form of altered intercellular communication. This should be added.

Response: We added the new sentence in the Introduction section.

5. The suggestion to include some discussion and critical remarks has not been taken up by the authors. Reviewing merely findings provided in literature is not what a review should do, it should provide context and guidance to the readership.

Response: the intent of our review was to provide an overview of the most significant available data about cellular and animal models involving EVs in the COPD pathogenesis.  We believe that critical remarks are more appropriate in the discussion of original research papers. However, in the manuscript production we have tried to give a critical view of the revised literature: in this light, along with the conclusion paragraph, we have added some new sentences in the three main paragraphs revising study on EVS in peripheral blood, in BALF and in cellular/animal models of COPD, in order to better clarify both the potential and the criticisms of EVs in COPD.

6. I repeat my comment on the use of neutrophil EVs vs other markers of neutrophils - EVs are not easily measured, lots of pitfalls.

Response:  We agree with the reviewer that there are more specific markers for neutrophils, but our intent was to emphasize that neutrophils communicate with other cells by releasing EVs. MVs represent an inflammatory blueprint of the activation status of various cells (neutrophils included), and not mere cell markers. In fact, in the paper of Genschmer (ref 51) and also in that of Soni (ref 52) the EVs released by neutrophils are not to be considered as markers of these cells, but a way of neutrophils communication. This role of EVs is the same that they have for other cell types, besides neutrophils.

7. Secondly, it should at least be mentioned that although in health epithelial cells represent a major source of EVs, this has not been examined or found in clinical samples in COPD. Can the authors speculate on the reasons? This is especially relevant given that in vitro studies do focus on these cells as source, but also in relation to the spread of senescence from senescent epithelial cells.

Response: we modified the text by removing the sentence.

8. There is a lack of connection between the clinical evidence and some of the mechanistic studies and conceptual framework that deserves attention.

Response: We agree with the reviewer that the connection between the clinical evidence and some of the mechanistic studies is missing. We have therefore tried to make it clearer in paragraph 3.3, even though the revised studies have been conducted on cellular/animal models, and not on COPD patients, thus limiting their potential translation to COPD in humans.

As shown by the number of publications, there has been an exponential increase in the interest in EVs and their potential applications in understanding the underlying mechanisms of various diseases, such as cancer, cardiovascular, metabolic, neurological, and infectious diseases, among others, which have revealed a role for EVs as promising biomarker candidates for diagnosis, prognosis, and even therapeutic tools, in lung and other diseases. We think that mechanistic studies on EVs in the pathogenesis of COPD are developing and this is an important and interesting field that needs to be further explored; unfortunately, current evidences do not allow firm conclusions with regard to COPD in humans.

9. The addition of the figure is a good start - but it does not include COPD or the blood. Instead it contains info that is not mentioned in the text (markers of cellular origin of EVs, platelets as a source).

Response: As requested by reviewer 1 in the first revision, we added a figure to emphasise the origins of EVs in COPD and their surface markers. We have now modified the figure, as requested by the reviewer.